# Assessment of heavy metal pollution in surface sediments of the Chishui River Basin, China

**Fanxi Li** [1], **Xia Yu** [2,3,4]*, **Jiemei Lv** [2], **Qixin Wu** [3], **Yanling An** [2,3]*

**1** Key Laboratory of Karst Environment and Geohazard Prevention, Guizhou University, Guiyang, Guizhou Province, China, **2** The College of Resources and Environmental Engineering, Guizhou Institute of Technology, Guiyang, Guizhou Province, China, **3** The College of Resources and Environmental Engineering, Guizhou University, Guiyang, Guizhou Province, China, **4** State Key Laboratory of Loess and Quaternary Geology, Institute of Earth Environment, Chinese Academy of Sciences, Xi'an, Shaanxi Province, China

* yuxia@git.edu.cn (XY); anyanling@git.edu.cn (YA)

**Data Availability Statement:** All relevant data are within the paper and its Supporting Information files.

**Funding:** This study was funded by the Open Fund of the State Key Laboratory of Loess and Quaternary Geology [grant number SKLLQG2036],

## Abstract

Accumulated heavy metals in surface sediments are released into the aquatic environment, causing secondary contamination of the hydrosphere, and increasing the risks to human health. To evaluate the pollution characteristics of heavy metals in the sediments of the Chishui River Basin, in the present study, the concentrations of five heavy metals in surface sediments of the Chishui River Basin in China were investigated using the geo-accumulation index, pollution load index, and potential ecological risk indexes. These indexes evaluated the degree of contamination and the influence of human activities on heavy metal levels in the basin. Cu, Zn, Cd, Hg, and As were found at concentrations of 5.12–120.40, 36.01–219.31, 0.03–1.28, 0.01–1.18, and 1.56–11.59 mg kg$^{-1}$, respectively, with mean values of 37.43, 91.92, 0.25, 0.07, and 5.16 mg kg$^{-1}$, respectively, in the order Zn > Cu > As > Cd > Hg. The contamination indices revealed Hg as the principal pollutant based on the spatial distribution, while Pearson's correlation coefficients suggested that Cu, Zn, and As originated from a similar source. Hg had a different source from the other metals, whereas Cd originated from a different source compared with that of Zn, As, and Hg. This paper showed a Hg and Cd contamination in the Chishui River Basin.

## Introduction

Contaminants containing high concentrations of heavy metals continue to be discharged into aquatic systems. These metals are often deposited on the bottom of such systems via precipitation and flocculation, thereby transforming the associated sediments into heavy metal repositories [1–3]. Due to their non-degradability, toxicity, and resistance to metabolization [4,5], heavy metals in sediments can harm aquatic organisms, as well as human health, through bioaccumulation and bioamplification [6]. In aquatic ecosystems, the proportion of heavy metals present as dissolved ions is low because most metals are deposited in the associated sediments [4,7,8]. However, heavy metals in sediments can be released and discharged into

the Chinese Academy of Sciences Strategic Leading Science and Technology Project (Category B) [grant number XDB40000000], the Guizhou Institute of Technology High-level Talent Research Startup Funding Project [grant number 2019, 45], the Science and Technology Program of Guizhou Province (Guizhou Science and Technology Cooperation Support [grant number 2018, 20101]), and the Science and Technology Foundation of Water Resources Department of Guizhou Province [grant number KT201508]. The funder 1(Yanling An) provided financial assistance in our study design, data collection and analysis, funder 2(Peng Cheng) helped us analyze the articles and play a significant role in the polishing of this paper.

**Competing interests:** The authors have declared that no competing interests exist.

aquatic systems via changes in water conditions, such as the hydrodynamics, temperature, and pH, causing secondary pollution [9].

Assessment of sediment heavy metal pollution is critical for the ecological protection of the Chishui River. The average water volume in the Chishui River in the past few years has been reported to be $9.74 \times 10^9 \, m^3$ while the average flow rate into estuaries is approximately 309 $m^3$/s [10]. The Chishui River and other tributaries are naturally connected to the Yangtze River, and the associated hydrological processes provide suitable breeding conditions for migratory fish species [11]. According to Wu (2010), among the 135 fish species reported in the Chishui River, approximately 40 were considered endemic to the upper Yangtze in 2007 [11]. Therefore, the survival of these species is threatened by the input of contaminants, such as heavy metals and organic pollutants, into the ecosystem.

Previous studies have focused on the development and protection of cultural [12], touristic [13], and natural [11,14,15] resources along the Chishui River. Among the few available studies on the physicochemical properties of the Chishui River, Wu (2001) systematically identified the major components of the river water, as well as the background composition and characteristics of 15 trace elements in the raw and filtered water, suspended solids, sediments, and aquatic organisms, it is found that the total amount of chalocophile elements is lower than siderophile elements in Chishui River. Additionally [16], Zou et al. (2010) and Ji et al. (2012) investigated total phosphorus, suspended solids, pH values and biochemical oxygen demand, ammonia as indicators, separately, to analyzed the water quality variations in Chishui River [17,18], the water quality results showed a large fluctuation in the middle stream of Chishui River and it may be polluted by domestic sewage from the surrounding tributaries. Furthermore, Jiang et al. (2013), Lv et al. (2013) and Luo et al. (2014) also investigated the compositional variations along the Chishui River during the dry season, the results display that the ion content in Chishui River was affected by human activities such as agriculture and fossil fuel burning [19–21]. Thus, limited studies have examined heavy metal pollution and their spatial distributions in the sediments along the entirety of the Chishui River Basin [11].

Therefore, the principal objective of this study was to determine the concentrations of five heavy metals, i.e., Cu, Zn, Cd, Hg, and As, in the surface sediments in the Chishui River, China. The extent of pollution due to these metals was characterized using the geo-accumulation, pollution load index (PLI), and potential ecological risk indexes (ERI). The findings of this study may be useful for future investigations on heavy metals in river ecosystems, heavy metal pollution management, and policy formulation.

## Materials and methods

### Sediment sample collection

The Chishui River (104°45′–106°51′ E, 27°20′–28°50′ N) in southwest China is located at the transitional zone between the Yunnan-Guizhou Plateau and the Sichuan Basin (Fig 1). Although no dam has been constructed on the river, it is an important tributary along the upper reaches of the Yangtze River. It flows through 13 counties in three provinces, with a mainstream length of 436.5 km, a natural head of 1,580 m, and a basin area of $1.91 \times 10^4 \, km^2$ [15]. Totally, 32 sediment samples were collected in the whole Chishui River basin, 19 sites were on the main stream and the other 13 sites were on the tributaries, which basically covered the entire basin of the Chishui River. 1–8 sampling sites were located at upstream, sites 9–16 were distributed in the middle stream and sites 17–24 were at downstream.

Samples were collected during a dry period in December 2012. At each site, sediment layers from the top 0–10 cm were collected from various points, which were mixed to produce a composite sample. Sediment samples were collected from 32 stations (Fig 1) throughout the

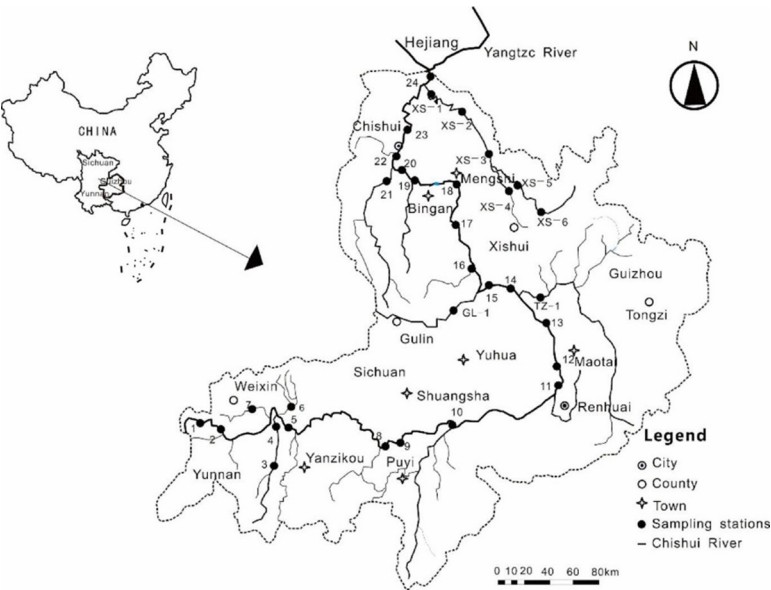

**Fig 1. Map of the sampling stations in the Chishui River Basin, China.**

Chishui River Basinand their coordinates of all samples were listed in Table 1. The samples were sealed in plastic bags, stored at 4˚, and transported to the laboratory for heavy metal analyses. In the laboratory, the sediments were spread on plastic films; stones, branches, and other plant materials were removed. The samples were then stored under dry conditions at room temperature. After gently rolling using a wooden stick according to the four-diagonal method [21], samples smaller than 200-mesh were collected and stored in polyethylene bags for testing.

## Heavy metal analyses

All analyses were performed at the Key Laboratory of Karst Environment and Geohazard Prevention, Ministry of Education, Guizhou University. An $HNO_3$-HF mixture was added to approximately 0.05 g of sediment in a Teflon vessel, and the mixture was subjected to digestion at 140˚C on a hot plate. The sample was then removed from the acid mixture after it appeared white or light-colored. Cu, Zn, and Cd concentrations in the sediments were determined using flame atomic absorption spectrometry (AAS; Contr AA 700, Germany) while Hg and As concentrations were measured via cold vapor AAS using the digested sample (0.3 g; GB/T 22105.1–2008). To ensure accuracy and precision of the measurements, stream sediment (GBW07309) and soil (GBW07401) standards were used for quality control. Samples were analyzed in triplicate, and the relative standard deviations were < 5%. Ultrapure water was used for sample preparation for all tests, and all reagents were of guaranteed quality.

The SPSS Statistics software (version 25.0, IBM) was used for analyzing the correlation matrix of the heavy metals present in the surface sediments. Pearson correlation matrix were calculated for measured elements separately to identify the similarities, a $p$-value of $< 0.05$ was taken as significant. Microsoft Excel (version 2019) was used for statistical analysis of the test data.

## Sediment pollution assessment methods

**Geo-accumulation index.** $I_{geo}$ is an effective parameter for assessing heavy metal pollution levels in sediments. It can be obtained using the following equation:

$$I_{geo} = \log_2[C_n/(1.5 \times B_n)] \tag{1}$$

Table 1. The coordinates of sampling sites of the Chishui River.

| Site name | Latitude (deg and min N) | Longitude (deg and min E) |
|---|---|---|
| 1 | 27°41.958′ | 105°3.738′ |
| 2 | 27°42.124′ | 105°6.102′ |
| 3 | 27°43.163′ | 105°12.773′ |
| 4 | 27°44.951′ | 105°12.197′ |
| 5 | 27°43.736′ | 105°16.029′ |
| 6 | 27°47.431′ | 105°15.731′ |
| 7 | 27°46.790′ | 105°7.576′ |
| 8 | 27°39.705′ | 105°37.778′ |
| 9 | 27°40.715′ | 105°41.866′ |
| 10 | 27°43.735′ | 105°55.974′ |
| 11 | 27°48.798′ | 106°19.210′ |
| 12 | 27°52.350′ | 106°19.727′ |
| 13 | 27°57.427′ | 106°19.170′ |
| 14 | 28°8.770′ | 106°10.436′ |
| 15 | 28°9.483′ | 106°5.229′ |
| 16 | 28°14.600′ | 106°0.159′ |
| 17 | 28°21.005′ | 105°57.483′ |
| 18 | 28°29.464′ | 105°54.615′ |
| 19 | 28°29.022′ | 105°45.829′ |
| 20 | 28°31.556′ | 105°43.273′ |
| 21 | 28°30.321′ | 105°40.919′ |
| 22 | 28°33.217′ | 105°40.857′ |
| 23 | 28°37.113′ | 105°43.954′ |
| 24 | 28°48.190′ | 105°49.307′ |
| XS-1 | 28°45.713′ | 105°50.212′ |
| XS-2 | 28°40.876′ | 106°0.398′ |
| XS-3 | 28°33.396′ | 106°5.189′ |
| XS-4 | 28°28.960′ | 106°7.794′ |
| XS-5 | 28°29.914′ | 106°11.239′ |
| XS-6 | 28°24.188′ | 106°18.614′ |
| TZ-1 | 28°7.562′ | 106°19.645′ |
| GL-1 | 28°6.445′ | 105°59.052′ |

where $C_n$ represents the concentration of the heavy metal n and $B_n$ represents the background level (mg kg$^{-1}$). A factor of 1.5 was used for lithological variations in the background value, based on previously reported values for shales [22]. The $I_{geo}$ classes established by Muller (1969) for heavy metal pollution are presented in Table 2.

**Pollution load index.** The pollution load index (PLI), proposed by Tomlinson (1980), is used to evaluate the overall toxicity status of a sample associated with heavy metals [23]. It reflects the changing trends in heavy metal pollution in time and space. The PLI can be calculated using the followed equations:

$$CF_i = C_i / B_i \tag{2}$$

$$PLI_{site} = \sqrt[n]{CF_1 \times CF_2 \times \cdots\cdots \times CF_n} \tag{3}$$

$$PLI_{zone} = \sqrt[m]{PLI_1 \times PLI_2 \times \cdots\cdots \times PLI_m} \tag{4}$$

**Table 2. Congruent relationships between metal $I_{geo}$ values and pollution levels [22].**

| $I_{geo}$ | Contamination level |
|---|---|
| $I_{geo} \leq 0$ | Unpolluted |
| $0 < I_{geo} \leq 1$ | Unpolluted to moderately polluted |
| $1 < I_{geo} \leq 2$ | Moderately polluted |
| $2 < I_{geo} \leq 3$ | Moderately to heavily polluted |
| $3 < I_{geo} \leq 4$ | Heavily polluted |
| $4 < I_{geo} \leq 5$ | Heavily to extremely polluted |
| $I_{geo} > 5$ | Extremely polluted |

where $CF_i$ represents a pollution coefficient, $C_i$ represents the measured concentration of a metal in the sediments, $B_i$ represents the background value of a heavy metal, n represents the number of heavy metals investigated, and m represents the number of sampling sites. Table 3 lists the classes of the PLI and their corresponding contamination levels.

**Potential ecological risk index.** The potential ecological risk index (ERI) is used to assess the level of heavy metal pollution in sedimentary environments [24]. This is a widely utilized advanced index, which investigates the heavy metal content, the ecological effect of heavy metals, environmental benefits, and toxicology. The potential ecological risk factor, $E_r^i$, can be calculated as follows:

$$E_r^i = T_r^i \times \frac{C^i}{C_n^i} \tag{5}$$

where $T_r^i$ represents the toxic-response factor of a given substance, $C^i$ represents the concentration of metal $i$ in the sediments, and $C_n^i$ denotes the background value of metal $i$. According to the findings of Hakanson (1980), the toxic response factors for Zn, Cu, As, Cd, and Hg, are 1, 5, 10, 30, and 40, respectively.

The ERI is calculated by summing the $E_r^i$ values, as follows:

$$ERI = \sum_1^n E_r^i \tag{6}$$

Background values (BV) for Hg, Cd, As, Cu, and Zn are 0.034, 0.15, 7.6, 21.5, and 73.6 mg kg$^{-1}$, respectively [25,26]. Table 4 lists the derived five categories of $E_r^i$ and four classes of ERI.

## Results and discussion

### Descriptive statistics for heavy metals

Concentration levels in sediments showed a variation with the distance from the start of the river (Fig 2). The heavy metal concentration level of upstream is higher than downstream, it indicated that there was no large-scale heavy metal pollution in downstream. Further, a point source pollution may appeal in tributaries as Tongzi River (TZ-1) was affected by As Pollution and Gulin River (GL-1) was polluted by Cd.

**Table 3. Pollution grading standards based on the pollution load index (PLI) [23].**

| PLI | Contamination level |
|---|---|
| PLI < 1.0 | No contamination |
| $1.0 \leq PLI < 2.0$ | Moderately contaminated |
| $2.0 \leq PLI < 3.0$ | Considerably contaminated |
| $PLI \geq 3.0$ | Strongly contaminated |

**Table 4. Indexes of the potential ecological risk and grades [24].**

| Potential Ecological risk factor ($E_r^i$) | Ecological risk level | Potential Ecological risk index (ERI) | Ecological risk level |
|---|---|---|---|
| Er < 40 | Low potential ecological risk | ERI ≤ 150 | Low potential ecological risk |
| 40 ≤ Er < 80 | Moderate potential ecological risk | 150 ≤ ERI < 300 | Moderate potential ecological risk |
| 80 ≤ Er < 160 | Considerable potential ecological risk | 300 ≤ ERI < 600 | Considerable potential ecological risk |
| 160 ≤ Er < 320 | High potential ecological risk | 600 ≤ ERI | Very high ecological risk |
| 320 ≤ Er | Very high ecological risk | | |

Fig 2 presents trend based on the metal concentrations in the sediments from the Chishui River Basin. The sum of the heavy metal concentrations (Cu, Zn, Cd, Hg, and As) in the 32 samples varied between 51.92 and 314.10 mg kg$^{-1}$, yielding an average total concentration of 134.84 mg kg$^{-1}$.

The concentrations of Cu, Zn, Cd, Hg, and As in the samples ranged from 5.12–120.40, 36.06–219.33, 0.03–1.28, 0.01–1.18, and 1.57–11.59 mg kg$^{-1}$, respectively, with corresponding mean values of 37.43, 91.93, 0.25, 0.07, and 5.16 mg kg$^{-1}$. The sum of the Zn and Cu concentrations represented 95.94% of the total heavy metal concentration, with the following order for the average sediment concentrations: Zn > Cu > As > Cd > Hg. The coefficient of variation values presented in Table 5 varied from 43% for As to 280.37% for Hg, producing the following sequence: Hg > Cd > Cu > Zn > As. These findings highlight the higher spatial variations for Hg, Cd, and Cu relative to Zn and As.

In this study, the Cu concentration in 65.63% of the samples exceeded the BV (Fig 2A). Overall, samples obtained from the upstream region showed higher Cu concentrations than those obtained from the middle and lower reaches of the Chishui River. The maximum Cu concentration (sample 7) was 5.6-fold higher than the BV while the minimum concentration (sample 19) was 24% of the BV. This is because site 7 is in Weixin County, where industrial effluents and other pollutants from human activities are common. Similarly, the maximum Zn concentration measured in sample 12 was approximately three-fold higher than the BV while the minimum value obtained from XS-1 was 49% of the BV. The concentrations of As in all samples were lower than those of Cu and Zn, but were higher than those of Cd and Hg (Fig 2C). The highest As concentration, obtained from TZ-1, was almost 1.5-fold higher than the BV while Cd associated with GL-1 was 8.53-fold greater than the BV. The high Cd levels in the GL-1 samples indicate severe Cd contamination in the Gulin River: the mining industry in this area discharges industrial wastewater into this river, and soils in the area are likely contaminated with Cd (Fig 2B). The highest Hg concentration was found in sample 24 collected from Hejiang County, which is located at a site where the Chishui River flows into the Yangtze River; its value was 39.33-fold greater than that of the BV. We note that Hg pollution in the Xishui River is not considerable, with no Hg accumulation observed upstream of site 24, suggesting that the Chishui River is severely impacted by the anthropogenic activities in Hejiang County during its flow into the Yangtze River. A comparison of the average metal concentrations in the collected samples with the BV reveals that the sediments are contaminated. According to previous studies [27,28], anthropogenic activities are responsible for the high metal concentrations in the sediments of the Chishui River.

A comparison of the data from this study with monitoring data from other areas in China [29–31] reveals relatively lower heavy metal concentrations in the study area (Fig 2). The concentrations of metals were lower than those of samples collected from the Tuojiang and East rivers, indicating that, owing to a reduced influence from industrialization in Guizhou Province relative to other parts of China, there is less riverine pollution.

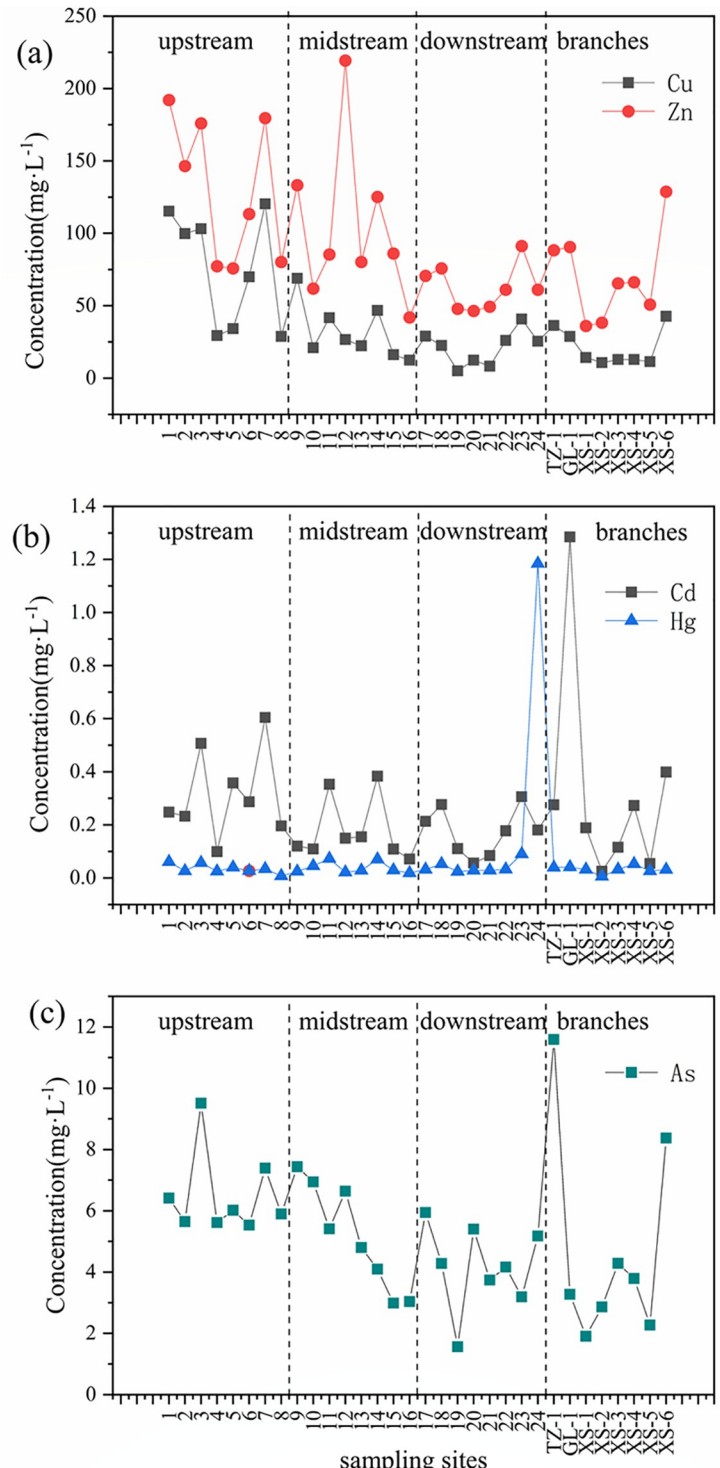

**Fig 2. Variation of measured heavy metals in the surface sediments of Chishui River Basin based on the direction of flow.**

**Table 5. Concentrations (mg kg$^{-1}$) of the five heavy metals in the sediments collected from the Chishui River compared to other rivers in China and the world.**

| | Cu | Zn | Cd | Hg | As | Reference |
|---|---|---|---|---|---|---|
| Concentration range | 5.12–120.40 | 36.01–219.31 | 0.03–1.28 | 0.01–1.18 | 1.56–11.59 | This study |
| Mean | 37.43 | 91.92 | 0.25 | 0.07 | 5.16 | -- |
| Median | 27.77 | 78.80 | 0.19 | 0.03 | 5.29 | -- |
| Coefficient of variation | 85.13% | 51.86% | 92.78% | 280.37% | 43% | -- |
| Background Value | 21.5 | 73.6 | 0.15 | 0.034 | 7.6 | [25] |
| Tuojiang | 48.95 | 261 | 1.48 | 0.19 | 11.84 | [29] |
| East River | 157.29 | 213.21 | 0.98 | 0.42 | —— | [30] |
| Mean concentration from rivers in China | 21 | 68 | 0.14 | 0.042 | 9.1 | [31] |
| Preindustrial reference value for lake sediments | 50 | 175 | 1.0 | 0.25 | 15 | [24] |
| Swarnamukhi River Basin | 100.9 | 63.4 | 0.2 | —— | —— | [32] |
| Halda River | 5.9 | 79.58 | 0.04 | 0.001 | —— | [33] |
| Thamirabarani River | 35.236 | 93.278 | 3.123 | —— | 2.061 | [34] |
| Hunza River | 8 | 27 | 0.4 | —— | —— | [35] |

Relative to the concentrations of heavy metals in rivers worldwide, Cu and Zn concentrations in the sediments from the Chishui River were correspondingly lower and higher than those of samples from the Swarnamukhi River Basin (India), whereas the Cd concentrations were comparable [32]. Conversely, the Cu, Zn, Cd, and Hg concentrations for samples collected from the Chishui River were higher than those of samples collected from the Halda River (Bangladesh) [33], with Cu and Cd concentrations approximately 6.3- and 6.2-fold higher, respectively. Additionally, sediments from the Chishui River exhibited lower Cd and higher As concentrations compared to those from the Thamirabarani River (India) [34]. Furthermore, the Cu and Zn concentrations of the sediments from the Chishui River were 4.6- and 3.4-fold higher than those of sediments from the Hunza River, respectively (Pakistan) [35]. These differences in the concentrations of heavy metals between the sediments in this study and those from global rivers can be attributed to the sampling sites, levels of contamination, regional characteristics, and anthropogenic activities [4] (Fabio et al., 2021).

## Sediment contamination assessment

**Geo-accumulation index assessment.** Table 6 presents the results of the $I_{geo}$ assessment, which highlights the extent of pollution associated with various metals. The $I_{geo}$ values for the elements ranged from -2.66 to 1.90 for Cu (mean = -0.23), -1.61 to 0.99 for Zn (mean = -0.43), -3.11 to 2.51 for Cd (mean = -0.28), -3.24 to 4.54 for Hg (mean = -0.5), and -2.86 to 0.02 for As (mean = -1.28). The negative mean $I_{geo}$ values for all elements indicate unpolluted areas. The $I_{geo}$ value for Hg from sampling site 24 is an outlier, whereas 18.75 and 9% of the sampling sites showed Cu and Cd accumulation, respectively. Moreover, upstream of the Chishui River, all heavy metals, except for As, were characterized by accumulation. The $I_{geo}$ values reported in the present study for Cu, Zn, and Cd were lower than Taihu lake [36] and Cu, Zn, Cd, As were lower than Longjiang River [37] and Xiaoqing River [38]. Compared with other rivers in China, the heavy metals in sediment of Chishui River were less polluted. The accumulation of the examined elements had the following order: Cu > Cd > Zn > Hg > As.

**Pollution load index assessment.** The PLI values of the 32 samples from the Chishui River Basin are shown in Fig 3. The values ranged from 0.31–2.47, with a median value of 1.09. According to these values, 9.37% of the samples were considerably contaminated, 43.75% were moderately contaminated, and 46.88% were uncontaminated. The considerably contaminated samples were concentrated upstream of the Chishui River (samples 1, 3, and 7); among these,

**Table 6.** $I_{geo}$ values for five heavy metals in Chishui River sediments.

| Sample No. | $I_{geo}$ | | | | |
|---|---|---|---|---|---|
| | **Cu** | **Zn** | **Cd** | **Hg** | **As** |
| **1** | 1.84 | 0.80 | 0.14 | 0.25 | −0.83 |
| **2** | 1.63 | 0.41 | 0.05 | −0.96 | −1.01 |
| **3** | 1.68 | 0.67 | 1.17 | 0.17 | −0.26 |
| **4** | −0.14 | −0.51 | −1.18 | −1.01 | −1.02 |
| **5** | 0.08 | −0.54 | 0.67 | −0.34 | −0.92 |
| **6** | 1.12 | 0.04 | 0.35 | −0.94 | −1.04 |
| **7** | 1.90 | 0.70 | 1.43 | −0.57 | −0.62 |
| **8** | −0.16 | −0.46 | −0.20 | −2.69 | −0.95 |
| **9** | 1.10 | 0.27 | −0.91 | −1.05 | −0.62 |
| **10** | −0.62 | −0.84 | −1.04 | −0.16 | −0.71 |
| **11** | 0.37 | −0.37 | 0.65 | 0.51 | −1.08 |
| **12** | −0.27 | 0.99 | −0.59 | −1.24 | −0.78 |
| **13** | −0.53 | −0.46 | −0.54 | −0.84 | −1.25 |
| **14** | 0.54 | 0.18 | 0.77 | 0.46 | −1.48 |
| **15** | −0.99 | −0.36 | −1.05 | −0.79 | −1.93 |
| **16** | −1.37 | −1.40 | −1.65 | −1.42 | −1.91 |
| **17** | −0.15 | −0.64 | −0.08 | −0.66 | −0.94 |
| **18** | −0.51 | −0.54 | 0.30 | 0.05 | −1.41 |
| **19** | −2.66 | −1.21 | −1.03 | −1.09 | −2.86 |
| **20** | −1.37 | −1.25 | −2.01 | −0.85 | −1.08 |
| **21** | −1.95 | −1.16 | −1.41 | −0.82 | −1.61 |
| **22** | −0.31 | −0.85 | −0.34 | −0.65 | −1.45 |
| **23** | 0.34 | −0.27 | 0.44 | 0.83 | −1.84 |
| **24** | −0.34 | −0.85 | −0.32 | 4.54 | −1.14 |
| **TZ-1** | 0.17 | −0.32 | 0.29 | −0.36 | 0.02 |
| **GL-1** | −0.16 | −0.29 | 2.51 | −0.31 | −1.80 |
| **XS-1** | −1.18 | −1.61 | −0.25 | −0.68 | −2.58 |
| **XS-2** | −1.58 | −1.53 | −3.11 | −3.24 | −1.99 |
| **XS-3** | −1.33 | −0.75 | −0.96 | −0.66 | −1.41 |
| **XS-4** | −1.32 | −0.74 | 0.28 | 0.06 | −1.59 |
| **XS-5** | −1.50 | −1.12 | −2.05 | −0.96 | −2.33 |
| **XS-6** | 0.40 | 0.22 | 0.83 | −0.69 | −0.44 |
| **mean** | −0.23 | −0.43 | −0.28 | −0.50 | −1.28 |

sample 3 showed the maximum PLI (Fig 3). Mean PLIs of 1.40, 0.96, and 0.85 corresponded to the upstream, midstream, and downstream portions of the Chishui River, respectively, demonstrating its efficient self-purifying potential. The PLI value of upstream was higher than river sediments from Wuhu (mean PLI = 1.24) [39] while other portions of Chishui River possessed a lower PLI value. The PLI values for the three branches had the following order: GL-1 (1.53) > TZ-1 (1.50) > XS1–X6 (0.8). The average PLI value of 1.8 for the entire basin indicates moderate contamination (PLI > 1).

**Potential ecological risk assessment.** The Er and ERI values for Cu, Zn, Cd, Hg, and As in the sediments from the Chishui River Basin are presented in Table 7 and Fig 4. The Er values ranged from 1.19–28 (mean = 8.79), 0.49–2.98 (mean = 1.25), 5.2–256.98 (mean = 50.04), 6.35–1393.29 (mean = 85.48), and 2.06–15.25 (mean = 6.79) for Cu, Zn, Cd, Hg, and As,

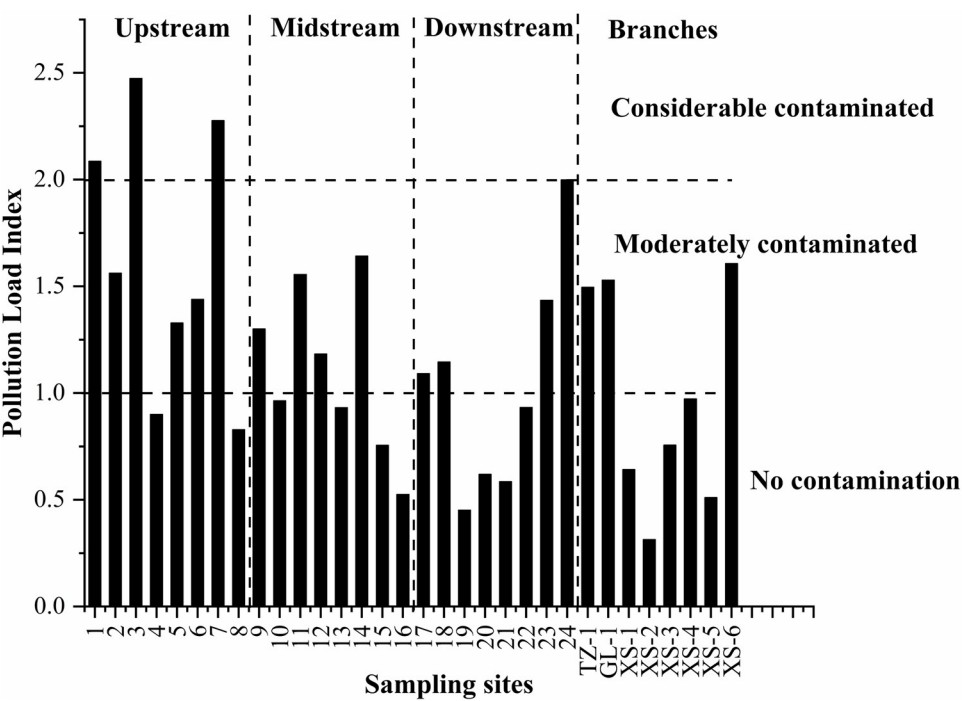

**Fig 3. Pollution load index for heavy metals in the sediments at different sites in the Chishui River Basin, China.**

respectively. The potential ecological risk associated with Hg was the highest among the heavy metals investigated in the surface sediments from the Chishui River Basin. The potential ecological risk index values ranged from 6.35 to 1,393.29, with a median value of 37.71, which suggests that most of the sampling sites have a low risk. Approximately 36.88% of the samples exhibited a moderate risk for Cd. However, the Er values of < 40 for Cu, Zn, and As for all 32 samples indicate a low ecological risk for these metals. Overall, the mean Er values for the five heavy metals in the sediments had the following order: Hg > Cd > Cu > As > Zn. The high ecological risk linked to sample 24 in Table 7 is mainly attributed to Hg. Although site 24 likely reflects point source pollution, data for 75% of the investigated samples indicate a low ecological risk.

The five heavy metals produced ERI values ranging from 18.34 to 1,443.02 (mean = 152.26), which represent low to very high ecological risks. The ERI value of > 600 for sample 24 indicates a very high ecological risk site (the ERI value of 1,443.02 for sample 24 showed in Fig 4). The ERI value for the 14 sites in the GL-1 section was > 300, which represents a considerable ecological risk; six of the 32 ERI values were 150 ≤ ERI ≤ 300, denoting moderate ecological risk; and 24 of the 32 ERI values were < 150, implying a low ecological risk. Hg contributes significantly to the ERI values because of its high toxicity and point source pollution, such as at site 24. Compared with other river in China, Chishui river has a lower ERI value than Tuo river(mean ERI = 198.31) [29] and Xiaoqing River(mean ERI = 173.31) [38], it means that Chishui River was less polluted. The samples with high ERI value were all distributed in the upper reaches of the Chishui River. The upper reaches of the Chishui River belong to the Yunnan-Guizhou plateau, and the vegetation coverage is lower than that of the lower reaches. Meanwhile, the agriculture is mainly sloping farmland, and the soil erosion is serious. Therefore, the higher upstream risk may be mainly influenced by primitive sloping farming practices and relatively high natural erosion.

**Table 7. Risk factors ($E_r^i$ values) for heavy metals in sediments from the Chishui River Basin, China.**

| Site | $E_r^i$ | | | | |
|------|------|------|--------|----------|-------|
| | Cu | Zn | Cd | Hg | As |
| 1 | 26.81 | 2.61 | 49.58 | 71.53 | 8.44 |
| 2 | 23.22 | 1.99 | 46.54 | 30.82 | 7.43 |
| 3 | 24.00 | 2.39 | 101.36 | 67.53 | 12.52 |
| 4 | 6.83 | 1.05 | 19.82 | 29.88 | 7.39 |
| 5 | 7.95 | 1.03 | 71.58 | 47.29 | 7.92 |
| 6 | 16.27 | 1.54 | 57.40 | 31.29 | 7.28 |
| 7 | 28.00 | 2.44 | 120.84 | 40.35 | 9.73 |
| 8 | 6.71 | 1.09 | 39.30 | 9.29 | 7.76 |
| 9 | 16.04 | 1.81 | 24.02 | 28.94 | 9.79 |
| 10 | 4.89 | 0.84 | 21.86 | 53.76 | 9.14 |
| 11 | 9.71 | 1.16 | 70.62 | 85.29 | 7.12 |
| 12 | 6.21 | 2.98 | 29.96 | 25.41 | 8.74 |
| 13 | 5.21 | 1.09 | 31.00 | 33.53 | 6.32 |
| 14 | 10.88 | 1.70 | 76.68 | 82.71 | 5.39 |
| 15 | 3.78 | 1.17 | 21.74 | 34.59 | 3.93 |
| 16 | 2.91 | 0.57 | 14.36 | 22.35 | 4.00 |
| 17 | 6.76 | 0.96 | 42.64 | 38.00 | 7.82 |
| 18 | 5.26 | 1.03 | 55.36 | 62.00 | 5.63 |
| 19 | 1.19 | 0.65 | 22.10 | 28.24 | 2.06 |
| 20 | 2.90 | 0.63 | 11.20 | 33.29 | 7.11 |
| 21 | 1.94 | 0.67 | 16.88 | 33.88 | 4.92 |
| 22 | 6.06 | 0.83 | 35.52 | 38.24 | 5.48 |
| 23 | 9.50 | 1.24 | 61.24 | 106.35 | 4.20 |
| 24 | 5.94 | 0.83 | 36.16 | 1,393.29 | 6.81 |
| TZ-1 | 8.45 | 1.20 | 55.16 | 46.59 | 15.25 |
| GL-1 | 6.71 | 1.23 | 256.98 | 48.47 | 4.31 |
| XS-1 | 3.32 | 0.49 | 37.74 | 37.53 | 2.51 |
| XS-2 | 2.50 | 0.52 | 5.20 | 6.35 | 3.77 |
| XS-3 | 2.99 | 0.89 | 23.14 | 37.88 | 5.64 |
| XS-4 | 3.01 | 0.90 | 54.64 | 62.47 | 4.99 |
| XS-5 | 2.65 | 0.69 | 10.86 | 30.94 | 2.99 |
| XS-6 | 9.92 | 1.75 | 79.74 | 37.29 | 11.02 |

## Discrepancy in different evaluation methods

Different evaluation methods were used to evaluate the pollution of heavy metals in the sediments of the Chishui River. The $I_{geo}$ values revealed that 5 heavy metals were unpolluted, the evaluation result of potential ecological risk index method shows that Cd and Hg was in moderate and considerable potential ecological risk, respectively. Although the two evaluation methods are calculated based on the soil environmental background value, the $I_{geo}$ is based on the environmental geochemistry, and the calculation results focus on reflecting the degree of pollution of heavy metals by human activities; While the potential ecological risk index method is to calculate the potential ecological risk factor of a single heavy metal from the perspective of the biological toxicity of heavy metals. The toxicity coefficient of Cd is 3 to 30 times that of other heavy metals, and a low concentration of Cd can cause huge damage to biological health. Therefore, the difference in toxicity coefficient greatly affects the evaluation results, leading to obvious discrepancy in the evaluation results of the two methods.

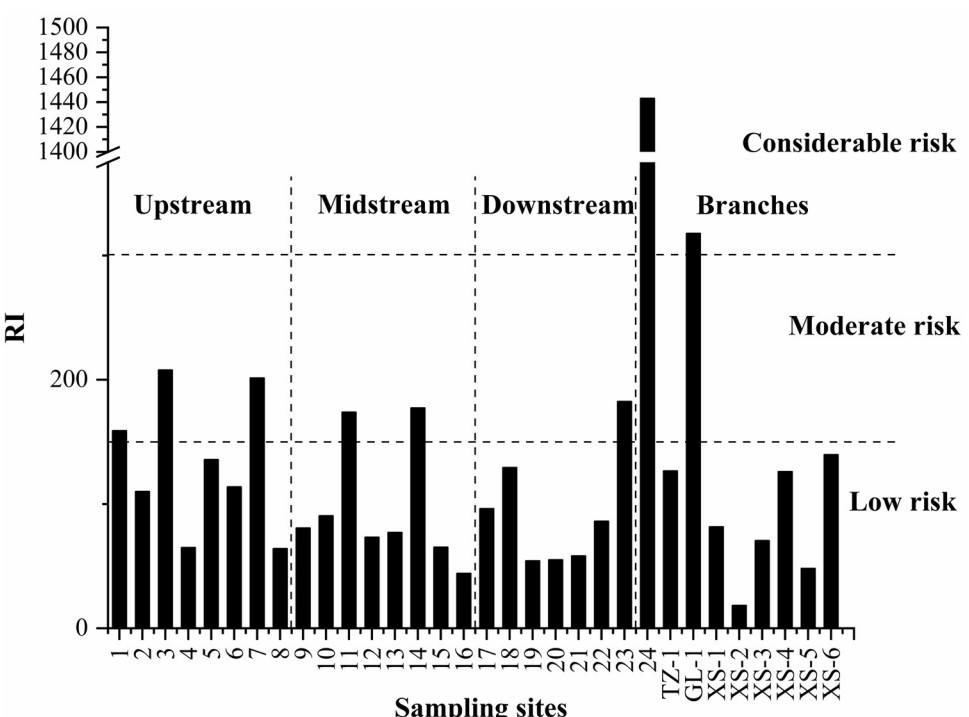

**Fig 4. Potential ecological risk indexes for heavy metals in sediments from the Chishui River Basin, China.**

## Heavy metal source apportionment

The Pearson correlation matrix is useful for determining the sources and pathways of contaminants in river surface sediments [1]. A correlation matrix for the elements studied is presented in Table 8. The confidence levels between Cu, Zn, and As were up to 99%, which suggests similar pollution sources for these heavy metals. The weak positive correlations between Cd and the other heavy metals (except Cu) indicates that Cd is likely associated with contaminant sources different from those of Hg, Zn, and As. Moreover, the weak negative correlations between Hg and the other metals (excluding As) imply no relationships among these metals [40]. In combination with ERI, the high-risk points were located near towns, indicating that metal ion content in Chishui River was seriously affected by human activities, Cu, Zn, and As were most likely derived from the discharge of industrial wastewater. Except for a few

**Table 8. Pearson correlation coefficient matrix for the heavy metals in the Chishui River surface sediments.**

|  | **Cu** | **Zn** | **Cd** | **Hg** | **As** |
|---|---|---|---|---|---|
| Cu | 1 |  |  |  |  |
| Zn | 0.796** | 1 |  |  |  |
| Cd | 0.351* | 0.324 | 1 |  |  |
| Hg | −0.043 | −0.098 | −0.023 | 1 |  |
| As | 0.538** | 0.558** | 0.154 | 0.009 | 1 |

Notes

** Correlation is significant at the 0.01 level (2-tailed)

* Correlation is significant at the 0.05 level (2-tailed).

sampling points, heavy metal pollution in the whole Chishui River basin has little impact on human beings.

## Conclusions

According to the $I_{geo}$ values, 18.75 and 9% of the sampled sites displayed Cu and Cd accumulation, respectively. The average PLI value for the entire basin indicates moderate contamination. The $E_r^i$ values for the five heavy metals followed the order Hg > Cd > Cu > As > Zn, with 24 samples considered low Er, six being moderate Er, one being considerable Er, and one being very high Er. Hg and Cd contributed significantly to the ERI values because of its higher toxicity.

Overall, the Chishui River Basin is characterized by moderate contamination. In the entire basin, relatively high risk sites are usually located in the upstream. The main reason may be the impact of agricultural farming and natural weathering of rock formations in the upper reaches of the basin. This study provides a reference for the formulation of policies in Guizhou. As the water source for Guizhou's wine industry, Chishui River is slightly polluted. In addition, Hg and Cd pollution in the Chishui River should be considered a serious problem.

## Supporting information

**S1 File. Highlights.**
(DOCX)

## Acknowledgments

We would like to thank graduate students, in particular Hao Jiang and Jin Luo, who provided assistance with sample collection and processing.

## Author Contributions

**Conceptualization:** Xia Yu, Jiemei Lv.

**Formal analysis:** Qixin Wu.

**Supervision:** Yanling An.

**Writing – original draft:** Fanxi Li.

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
