## [Decision Letter · Decision Letter 0]

14 Sep 2021

PONE-D-21-22502Assessment of Heavy Metal Pollution in Surface Sediments of the Chishui River Basin, ChinaPLOS ONE

Dear Dr. li,

Thank you for submitting your manuscript to PLOS ONE. After careful consideration, we feel that it has merit but does not fully meet PLOS ONE’s publication criteria as it currently stands. Therefore, we invite you to submit a revised version of the manuscript that addresses the points raised during the review process.

We look forward to receiving your revised manuscript.

Kind regards,

Xiaoshan Zhu, Ph.D.

Academic Editor

PLOS ONE

Journal Requirements:

2. In your Methods section, please provide additional location information of the sampling stations, including geographic coordinates for the data set if available.

3. In your Methods section, please provide additional information regarding the permits you obtained for the work. Please ensure you have included the full name of the authority that approved the sampling sites access and, if no permits were required, a brief statement explaining why.

This study was funded by the Open Fund of the State Key Laboratory of Loess and Quaternary Geology [grant number SKLLQG2036], the Chinese Academy of Sciences Strategic Leading Science and Technology Project (Category B) [grant number XDB40000000], the Guizhou Institute of Technology High-level Talent Research Startup Funding Project [grant number 2019, 45], the Science and Technology Program of Guizhou Province (Guizhou Science and Technology Cooperation Support [grant number 2018, 20101]), and the Science and Technology Foundation of Water Resources Department of Guizhou Province [grant number KT201508].

This study was funded by the Open Fund of the State Key Laboratory of Loess and Quaternary Geology [grant number SKLLQG2036], the Chinese Academy of Sciences Strategic Leading Science and Technology Project (Category B) [grant number XDB40000000], the Guizhou Institute of Technology High-level Talent Research Startup Funding Project [grant number 2019, 45], the Science and Technology Program of Guizhou Province (Guizhou Science and Technology Cooperation Support [grant number 2018, 20101]), and the Science and Technology Foundation of Water Resources Department of Guizhou Province [grant number KT201508].

This study was funded by the Open Fund of the State Key Laboratory of Loess and Quaternary Geology [grant number SKLLQG2036], the Chinese Academy of Sciences Strategic Leading Science and Technology Project (Category B) [grant number XDB40000000], the Guizhou Institute of Technology High-level Talent Research Startup Funding Project [grant number 2019, 45], the Science and Technology Program of Guizhou Province (Guizhou Science and Technology Cooperation Support [grant number 2018, 20101]), and the Science and Technology Foundation of Water Resources Department of Guizhou Province [grant number KT201508].

7. Please amend the manuscript submission data (via Edit Submission) to include author Xia Yu, Jiemei Lv, Qixin Wu and Yanling An.

8. We note that Figure 1 in your submission contain [map/satellite] images which may be copyrighted. All PLOS content is published under the Creative Commons Attribution License (CC BY 4.0), which means that the manuscript, images, and Supporting Information files will be freely available online, and any third party is permitted to access, download, copy, distribute, and use these materials in any way, even commercially, with proper attribution. For these reasons, we cannot publish previously copyrighted maps or satellite images created using proprietary data, such as Google software (Google Maps, Street View, and Earth). For more information, see our copyright guidelines: http://journals.plos.org/plosone/s/licenses-and-copyright.

Additional Editor Comments:

To meet the requires for publishing, the introduction, method, results and conclusion of the MS all need major revision.

Reviewers' comments:

Reviewer's Responses to Questions

**Comments to the Author**

1. Is the manuscript technically sound, and do the data support the conclusions?

Reviewer #1: Yes

Reviewer #2: No

2. Has the statistical analysis been performed appropriately and rigorously? 

Reviewer #1: Yes

Reviewer #2: No

3. Have the authors made all data underlying the findings in their manuscript fully available?

Reviewer #1: Yes

Reviewer #2: Yes

4. Is the manuscript presented in an intelligible fashion and written in standard English?

Reviewer #1: Yes

Reviewer #2: Yes

5. Review Comments to the Author

Reviewer #1: The primary goal of the manuscript was to examine the presence of heavy metal contaminants in surface sediments of the Chishui River Basin, China. The authors have used relevant methods and technology to conduct their research and their findings support their conclusions.

In page #6, the authors should verify the equation:   = log2[ /(1.5 × )]. If necessary,“i” may be replaced with “n”.

 It appears that the authors wanted to cite specific references for each equation but it is not in proper format. Please see below for an example and edit as needed.   Page#6:   = log2[ /(1.5 × )],     (1)

 Figure 2 (a,b,c) should be compiled for one figure (Page 12 and 13) 

The authors missed inserting line numbers in the whole manuscript.

The authors should write completely for different abbreviations. For example, in line # 3 of page 3 pollution load should be changed to “pollution load index. Similarly, RI should be replaced with ERI (line # 3 of Page #3). 

 1.      Is the manuscript technically sound, and do the data support the conclusions?

Yes, the authors have used relevant methods and technology to conduct their research and their findings support their conclusions. 

 2.      Has the statistical analysis been performed appropriately and rigorously?

 Yes, the authors have used appropriate statistical methods for data analyses.  

 3.      Have the authors made all data underlying the findings in their manuscript fully available? Yes, the authors have provided relevant data related to the manuscript.  

4.     Is the manuscript presented in an intelligible fashion and written in standard English?

The authors have written the manuscript in a logical fashion and used Standard English. The manuscript has a few typos and a few sentences should be rewritten.

Reviewer #2: The study evaluate the pollution characteristics of heavy metals in the sediments of the Chishui River Basin using five heavy metals in surface sediments and the geo-accumulation, pollution load, and potential ecological risk indexes. It is importance for the evaluation of environmental status and human being health. However, the data in the paper lack of replicates, which was not trustful enough. The introduction, method, results and conclusion all need major revision.

The specific comments can be found in the attached file.

6. PLOS authors have the option to publish the peer review history of their article (what does this mean?). If published, this will include your full peer review and any attached files.

Reviewer #1: No

Reviewer #2: No

---

## [Author Response · Author response to Decision Letter 0]

17 Oct 2021

Our respond has been uploaded in the form of files named “Cover letter” and “Response to Reviewers”

---

## [Decision Letter · Decision Letter 1]

15 Nov 2021

PONE-D-21-22502R1Assessment of Heavy Metal Pollution in Surface Sediments of the Chishui River Basin, ChinaPLOS ONE

Dear Dr. An,

Thank you for submitting your manuscript to PLOS ONE. After careful consideration, we feel that it has merit but does not fully meet PLOS ONE’s publication criteria as it currently stands. Therefore, we invite you to submit a revised version of the manuscript that addresses the points raised during the review process.

We look forward to receiving your revised manuscript.

Kind regards,

Xiaoshan Zhu, Ph.D.

Academic Editor

PLOS ONE

Journal Requirements:

Additional Editor Comments:

please address the reviewer's comments as soon as possible, thanks.

Reviewers' comments:

Reviewer's Responses to Questions

**Comments to the Author**

1. If the authors have adequately addressed your comments raised in a previous round of review and you feel that this manuscript is now acceptable for publication, you may indicate that here to bypass the “Comments to the Author” section, enter your conflict of interest statement in the “Confidential to Editor” section, and submit your "Accept" recommendation.

Reviewer #1: (No Response)

Reviewer #2: All comments have been addressed

2. Is the manuscript technically sound, and do the data support the conclusions?

Reviewer #1: Yes

Reviewer #2: Yes

3. Has the statistical analysis been performed appropriately and rigorously? 

Reviewer #1: Yes

Reviewer #2: Yes

4. Have the authors made all data underlying the findings in their manuscript fully available?

Reviewer #1: Yes

Reviewer #2: Yes

5. Is the manuscript presented in an intelligible fashion and written in standard English?

Reviewer #1: Yes

Reviewer #2: Yes

6. Review Comments to the Author

Reviewer #1: It appears that the authors have not edited the reference format for the equations (Line # 155, 168, 169, 170, 184, 190,

Many references are missing. For example, Wu (2001) (Line #71) and Zou et al (2010) (# 76). The authors should check carefully throughout the manuscript.

The authors should check carefully grammatical errors throughout the manuscript (e.g., line #108 – 112)

Reviewer #2: This paper evaluate the pollution characteristics of heavy metals in the sediments of the Chishui River Basin.

All the comments were well addressed. An accept is suggested.

7. PLOS authors have the option to publish the peer review history of their article (what does this mean?). If published, this will include your full peer review and any attached files.

Reviewer #1: No

Reviewer #2: No

---

## [Author Response · Author response to Decision Letter 1]

15 Nov 2021

We have revised the manuscript as required

---

## [Editor Report · Decision Letter 2]

19 Nov 2021

Assessment of Heavy Metal Pollution in Surface Sediments of the Chishui River Basin, China

PONE-D-21-22502R2

Dear Dr. An,

We’re pleased to inform you that your manuscript has been judged scientifically suitable for publication and will be formally accepted for publication once it meets all outstanding technical requirements.

Kind regards,

Xiaoshan Zhu, Ph.D.

Academic Editor

PLOS ONE
---

## [Editor Report · Acceptance letter]

24 Jan 2022

PONE-D-21-22502R2 

Assessment of Heavy Metal Pollution in Surface Sediments of the Chishui River Basin, China 

Dear Dr. An:

I'm pleased to inform you that your manuscript has been deemed suitable for publication in PLOS ONE. Congratulations! Your manuscript is now with our production department. 

Kind regards, 

on behalf of

Dr. Xiaoshan Zhu 

Academic Editor

PLOS ONE